# Incorporating Stereotactic Ablative Radiotherapy into the Multidisciplinary Management of Renal Cell Carcinoma

Rohit K. Raj [1], Rituraj Upadhyay [1], Shang-Jui Wang [1], Eric A. Singer [2] and Shawn Dason [2,*]

[1] Department of Radiation Oncology, The Ohio State University Comprehensive Cancer Center, Columbus, OH 43210, USA; rohit.raj@osumc.edu (R.K.R.); rituraj.upadhyay@osumc.edu (R.U.); shang-jui.wang@osumc.edu (S.-J.W.)
[2] Division of Urologic Oncology, The Ohio State University Comprehensive Cancer Center, Columbus, OH 43210, USA; eric.singer@osumc.edu
[*] Correspondence: shawn.dason@osumc.edu

**Abstract:** Stereotactic ablative radiotherapy (SABR) has challenged the conventional wisdom surrounding the radioresistance of renal cell carcinoma (RCC). In the past decade, there has been a significant accumulation of clinical data to support the safety and efficacy of SABR in RCC. Herein, we review the use of SABR across the spectrum of RCC. We performed an online search of the Pubmed database from January 1990 through April 2023. Studies of SABR/stereotactic radiosurgery targeting primary, extracranial, and intracranial metastatic RCC were included. For SABR in non-metastatic RCC, this includes its use in small renal masses, larger renal masses, and inferior vena cava tumor thrombi. In the metastatic setting, SABR can be used at diagnosis, for oligometastatic and oligoprogressive disease, and for symptomatic reasons. Notably, SABR can be used for both the primary renal tumor and metastasis-directed therapy. Management of RCC is evolving rapidly, and the role that SABR will have in this landscape is being assessed in a number of ongoing prospective clinical trials. The objective of this narrative review is to summarize the evidence corroborating the use of SABR in RCC.

**Keywords:** stereotactic ablative radiotherapy (SABR); stereotactic body radiotherapy (SBRT); radiation; renal cell carcinoma; oligometastasis; oligoprogression; IVC tumor thrombus





## 1. Introduction

Kidney cancers are amongst the top 10 cancers in the United States, with an estimated incidence of 81,000 cases and 14,890 attributable deaths in 2023 [1]. Malignant neoplasms of the kidney are complex, with several histologic types and distinct disease processes with variable clinical outcomes, the most prevalent of which is renal cell carcinoma (RCC), accounting for 80–85% of primary renal neoplasms [2]. Major subtypes of RCC include clear-cell (75–85%), papillary (10–15%), and chromophobe (5–10%) tumors. Other histological types of renal neoplasms are oncocytoma and angiomyolipoma (3–5%). At diagnosis, over one-fourth of patients present with regional and/or distant metastatic disease, and over half of patients eventually develop metastatic disease [3,4]. Overall survival across several prior studies ranges from 6 to 12 months in these patients with metastatic RCC (mRCC). While surgery remains the standard-of-care treatment modality for patients with localized RCCs, systemic therapy is the mainstay of treatment for mRCC. Historically, systemic therapies were limited to cytokine therapies, but more recently, several newer options have emerged, including immune checkpoint inhibitors (ICIs) and tyrosine kinase inhibitors (TKIs) [5–7].

In the context of management of localized, locally advanced, and metastatic RCC, stereotactic ablative radiotherapy (SABR) has emerged as another promising technique. Historically, RCC was considered to be a relatively radioresistant tumor, especially when

treated with conventional fractionation utilizing a dose of <5 Gy per fraction. Radio-resistance to conventionally fractionated RT was demonstrated by Deschavanne and Fertil using cell survival curves from in vitro studies [8]. However, more advanced radiation techniques have allowed the delivery of a higher dose per fraction to renal tumors safely, which results in more effective cell killing, as demonstrated by several recent studies in vitro as well as in vivo [9,10]. Although the underlying radiobiological phenomenon responsible for this effect is poorly understood, several hypotheses include alternate mechanisms of cell killing induced by SABR (vs. conventional RT) rather than mitotic catastrophe from DNA damage, including ceramide pathway cell killing, apoptosis, and vascular endothelial damage [11–13]. Another plausible explanation of the radiation response seen with SABR involves the immunostimulatory effects, especially in combination with immunotherapy [14]. Chow et al. evaluated the pathology specimens of patients with primary RCC treated with single-fraction SABR to a dose of 15 Gy followed by nephrectomy four weeks later, and observed broad transcriptional immune activation and clonality of immune cells within the tumor microenvironment [15].

SABR is generally defined as a radiotherapy technique in which high doses of radiation (>5–8 Gy per fraction) are administered to the target in 1–5 fractions with a high degree of anatomic accuracy [16]. Recent clinical studies have evaluated SABR for both primary and mRCC and have shown the safety as well as efficacy of this approach, with local control (LC) rates of 90–98% [17–19] and grade-3 toxicities attributable to SABR of less than 5% [20].

Current North American and European society guidelines recommend SABR as a subsequent line of local therapy for unresectable or medically inoperable primary RCC, or for the local treatment of metastatic sites in select patients with oligometastatic (OM) RCC or oligoprogression [21]. In this review, we aim to summarize the key contemporary evidence supporting the use of SABR in the settings of localized and metastatic RCC in the context of multimodality therapy with surgical resection and systemic therapies.

## 2. Evidence Acquisition

The medical literature including clinical trials, clinical studies, case reports, cohort studies, and review articles published from January 2000 to April 2023 was searched in PubMed. The search was performed on 1 May 2023 and did not have a language filter. Keywords used to search titles and abstracts included kidney cancer, renal cell carcinoma, RCC, advanced RCC, metastatic RCC, radiotherapy (RT), SBRT, SABR, and stereotactic radiosurgery (SRS). Any prospective or retrospective studies and review articles describing results with the use of SABR for RCC, with the original article in English or a translated version available, were included. Preclinical and modeling studies with clinical information and papers with incomplete, missing, or duplicated data were included. We did not require a minimum median follow-up duration for inclusion. The major objective of this descriptive review was to summarize the current literature supporting the use of SABR in the multidisciplinary management of localized and metastatic RCC, in combination with surgery and systemic therapy. We also aimed at highlighting the current knowledge gaps and the need for future prospective studies.

## 3. Primary Localized RCC

Most patients with localized RCC present with a renal mass noted incidentally on imaging for another indication [22]. Local symptoms of RCC include flank discomfort and hematuria. Patients with RCC are typically staged with imaging of the chest, abdomen, and pelvis [23]. Bone and brain metastases are uncommon in the absence of symptoms or lab abnormalities.

Management of the small renal mass (≤4 cm) is highly individualized [23]. Historically, most patients with a small renal mass were managed with surgical resection. A number of trends have emerged in the past decade to limit the morbidity of small renal mass management:

1. Renal mass biopsy. There is a high rate (31%) of benign pathology in resected renal masses [24]. An effort to reduce the morbidity of resecting benign renal masses has prompted increasing use of renal mass biopsy in patients with a small renal mass. A renal mass biopsy is diagnostic in 90% of small renal masses [25]. Utilization of biopsy prior to surgical resection is highly variable—it is likely most essential when a diagnosis other than renal cell carcinoma is suspected or biopsy results would impact the decision to proceed with treatment. A young patient unwilling to accept the uncertainties of a biopsy or a comorbid patient who would undergo active surveillance regardless of biopsy are good indications to avoid a biopsy [23]. Additional imaging with 99 mTc-sestamibi SPECT/CT [26] and 89Zr-DFO-girentuximab [27] are likely to become reasonable adjuncts to biopsy in the near future.

2. Active surveillance is also increasingly utilized for small renal masses. Intermediate-term outcomes with this approach are excellent. The metastatic rate is generally low (<1–2%) in well-selected patients [28]. The DISSRM registry is a prospective active surveillance registry that has recently reported on 585 patients with a 3.39 year median follow-up [29]. The rate of delayed intervention on active surveillance is 15%, with no difference in cancer-specific survival between those that elected primary vs. delayed intervention. The most common reasons for intervention are growth rate >0.5 cm/year or patient preference [30].

3. If treatment is elected, most patients are amenable to a robot-assisted partial nephrectomy. This has reduced morbidity over open partial nephrectomy and radical nephrectomy.

4. Other local therapies. percutaneous ablation offers a reasonable alternative to surgery and can be performed with cryoablation or radiofrequency ablation [31]. A biopsy should confirm malignancy before ablation. This approach has reduced morbidity relative to surgery, which may be particularly relevant to the patient with elevated surgical risk. LC rates may be slightly lower than that of a partial nephrectomy, which is relevant to a patient with a long life expectancy.

5. SABR. As discussed in the following sections, SABR is being increasingly considered.

Surgical candidates with a larger (>4 cm) renal mass generally undergo surgical resection with a partial or radical nephrectomy. A biopsy is less commonly performed given the higher rates of malignancy in this setting [32]. Ablative techniques [33], active surveillance [34], and SABR are alternatives to surgery that can be considered in a patient at elevated surgical risk.

### 3.1. SABR for Primary/Localized RCC

SABR is an emerging and effective alternative to surgery in patients who are medically inoperable or refuse surgery. In the mid-2000s, early clinical experiences with SABR for treatment of primary RCC were promising [15,35,36]. An early prospective dose escalation study of SABR by Sun et al. reported an overall LC rate of 92.7% with total SABR doses ranging from 21 to 48 Gy [37]. Furthermore, a small phase I trial of 19 patients showed that dose escalation to 48 Gy in four fractions was feasible without dose-limiting toxicities [38]. Subsequently, a phase 2 prospective trial of 37 patients demonstrated a 100% LC rate of localized RCC treated with SABR (26 Gy single fraction or 42 Gy in three fractions) at a median follow-up of 24 months with 3% grade 3 toxicity and no grade 4–5 toxicities [39].

Table 1 summarizes the studies reporting toxicity and/or outcomes with the use of SABR for localized and locally advanced RCCs. While the scale and number of prospective trials are limited, several retrospective studies further substantiate the efficacy of SABR for RCC. A recent systematic review and meta-analysis by Correa et al. demonstrated an excellent LC rate of 97.2% and low rates of grade 3–4 toxicity of 1.5%. Importantly, the authors reported a mean change in estimated glomerular filtration rate (eGFR) of only −7.7 mL/min following renal SABR, suggestive of the overall mild nephrotoxicity of therapy [40]. A large retrospective study of patients from the International Radiosurgery Oncology Consortium for Kidney (IROCK) conducted by Siva et al. also demonstrated the efficacy of SABR by reporting 4-year LC, cancer-specific survival (CSS), and progression-free survival

(PFS) of 97.8%, 91.9%, and 65.4%, respectively [41]. Longer-term follow-up continues to demonstrate efficacy and safety of SABR for localized RCC: individual patient data meta-analysis of IROCK patients found a 5.5% cumulative incidence of local failure at a median follow-up of 5 years, with only one patient developing grade > 3 toxicities (acute grade 4 duodenal ulcer and late grade 4 gastritis), and a median decrease in eGFR of 14.2 mL/min at 5 years [42]. The tolerability of SABR for treatment of RCC is further substantiated by several small series that treated frail patients with no report of grade ≥ 3 toxicity and preservation of quality of life [43–45].

SABR has also been shown to be an effective treatment for larger renal masses, namely ≥4 cm in size (≥T1b). In contrast, other non-surgical alternatives such as thermal ablation techniques are not suitable for ≥T1b disease due to an increased risk of local recurrence and complications [46]. Of note, it is important to keep in mind that SABR of a larger tumor, compared to that of a smaller tumor, is likely associated with a higher probability of cancer cell survival and lower rates of tumor control. For example, analyses of the IROCK cohort demonstrated an association with worsening PFS (HR 1.16, 95% CI 1.10–1.23; $p < 0.001$) and CSS (HR 1.28, 95% CI 1.19–1.39; $p < 0.001$) with increasing tumor size (per 1 cm increase in maximal dimension) [41]. Nevertheless, outcomes of SABR for larger RCC tumors are still quite favorable. In an analysis of 95 patients from the IROCK database with ≥T1b (>4 cm) primary RCC who received SABR, Siva et al. reported a 2.9% local failure rate and a PFS, OS, and CSS of 64.9%, 69.2%, and 91.4%, respectively, at 4 years post-treatment. Furthermore, there were no grade 3–5 toxicities post-treatment, with a modest post-SABR mean eGFR decrease of 7.9 mL/min [47]. In another study with 36 patients with RCC, of which 31 patients had ≥T1b disease, patients were treated with stereotactic magnetic resonance-guided RT, with a 1-year LC rate of 95.2% with no grade ≥ 3 toxicity [48].

**Table 1.** Studies summarizing outcomes with SABR for primary localized or locally advanced RCCs.

| Author (Year) | Study Type | No. of Patients (N) | Follow-Up Duration (Months) | Dose (Gy)/No. of Fractions | Grade 3+ Toxicity (%) | Local Control (%) | Comments, Study Population |
|---|---|---|---|---|---|---|---|
| Grelier et al. (2021) [44] | Retrospective | 23 | 22 | 35/5–7 | 0 | 96 | Frail patients unfit for surgery or other ablative therapies |
| Grubb et al. (2021) [49] | Prospective | 11 | 34.3 | 48/3 54/3 60/3 | 9.1 | 90 | Poor surgical candidates |
| Swaminath et al. (2021) [45] | Prospective | 28 | NA | 30–42/ 3–5 | NA | NA | 13 patients ≤ 4 cm, 19 with >4 cm tumors |
| Margulis et al. (2021) [50] | Prospective | 6 | 24 | 40/5 | 0 | NA | Neoadjuvant SABR for patients with IVC_TT |
| Tetar et al. (2020) [48] | Retrospective | 36 | 16.4 | 40/5 | 0 | 95.2 | MRI-guided SABR, 31 patients had ≥T1b disease |
| Siva et al. (2020) [47] | Retrospective | 95 | 32.4 | —— | 0 | 97.1 | Large (>4 cm), T1b or higher tumors |
| Senger et al. (2019) [43] | Retrospective | 10 | 27 | 24–25/1 36/3 | 0 | 92.3 | 7 with T1a and 3 with T3a disease |
| Hannan et al. (2023) [51] | Prospective | 16 | 36 | 36/3 or 40/5 | 0 | 94 (1-year) | ≤5 cm enlarging primary RCC |
| Siva et al. (2023) (FASTRACK II; Abstract only) [52] | Prospective | 70 | 42 | 26/1 (≤4 cm) 42/3 (>4 cm) | 10 | 100 | Non-surgical, T1b+ patients mostly |

### 3.2. SABR Dose Fractionation for Localized RCC

Several dose fractionation regimens have been used for treatment of localized RCC, but there is currently no high-level evidence to suggest an optimal dose for maximizing tumor control while minimizing toxicity. The systematic review and meta-analysis pub-

lished by Correa et al. used an array of dose fractionation schedules, the most common of which were 26 Gy in a single fraction and 30–45 Gy in 3–5 fractions. Local failures predominantly occurred in low-dose groups or with intentional underdosing of tumors to mitigate toxicity to adjacent organs [40]. In contrast, in a small prospective dose escalation study in 11 localized RCC patients who received 38, 54, or 60 Gy in three fractions, the authors reported an estimated 3-year LC rate of 90%, with only one patient (in the 60 Gy cohort) demonstrating local disease progression six months post-treatment without dose-limiting toxicity, suggesting that local failure can occur with high-dose regimens as well [49]. Of note, Siva et al. recently reported a significantly higher 5-year LC rate for single-fraction compared to multi-fraction SABR in their retrospective multivariable analysis of localized RCC patients in the IROCK database. However, the authors' analysis, with the assumption of an $\alpha/\beta$ value of 2.6 (belonging to A498, a common human RCC cell line), revealed a significantly lower biologically effective dose (BED) in the RCC patients treated with a range of multi-fraction SABR doses compared to those who received single-fraction SABR of 25 Gy, suggesting the possibility that an equivalently high BED with multi-fraction regimens could achieve similarly high LC. For this and other reasons, oncologic outcome comparisons based on SABR fractionation should be interpreted as exploratory and hypothesis-generating and should be evaluated through a prospective randomized trial [42]. The TransTasman Radiation Oncology Group 15.03 FASTRACK II phase 2 prospective trial is evaluating 26 Gy in one fraction for tumors $\leq$ 4 cm and 42 Gy in three fractions for tumors > 4 cm in size [53]. They recently presented their results (abstract only), demonstrating a local control of 100% after a median follow-up of 42 months, with 10% grade 3 toxicity and no grade 4 or higher toxicity.

### 3.3. Response Evaluation after SABR

Surveillance after SABR is most commonly done with computed tomography (CT) imaging using Response Evaluation Criteria in Solid Tumors (RECIST). However, because many RCC lesions are slow-growing, this may correspond to slower radiographic response post-SABR. Clinical experiences thus far indicate that post-SABR pseudoprogression is a possibility (initial increase in size, followed by regression), or stable disease initially post-SABR, followed by slow regression of tumor size over months and years [54]. As such, stable or slow regression in lesion size radiographically post-SABR is an expected and appropriate treatment response. Response evaluation is likely improved with multiparametric magnetic resonance imaging (MRI), as treatment effects can be detected earlier on diffusion and perfusion MRI, which correlates with subsequent tumor regression on CT [25].

## 4. Locally Advanced RCC

Renal cell carcinoma extending outside of the kidney without distant metastases is described as locally advanced. This may include involvement of the perinephric or renal sinus fat, renal vein or inferior vena cava (IVC), regional lymph nodes, or adjacent organs. In surgical candidates, locally advanced RCC is managed with surgical resection [55]. Adjuvant ICIs with pembrolizumab (anti-PD-1) for up to 1 year have been shown to reduce disease recurrence after nephrectomy by 37% for patients with locally advanced clear cell RCC [56,57]. Whether adjuvant pembrolizumab improves CSS has not yet been established.

### 4.1. Definitive Therapy with SABR for RCC with IVC Tumor Thrombus (IVC-TT)

It is estimated that ~10% of newly diagnosed RCC presents with disease invading the IVC and forming IVC-TT. RCC with IVC-TT portends a poor prognosis [58]. Furthermore, the presence of tumor thrombus can lead to several complications including venous congestion, Budd–Chiari syndrome, pulmonary embolism, or metastasis [59]. Non-metastatic RCC with IVC-TT is managed with radical nephrectomy and IVC thrombectomy. Medical inoperability, particularly for those with supradiaphragmatic tumor thrombus, is not uncommon in this population given the risks of surgery. SABR may be an effective alternative to surgery in this group. An early case report of two RCC patients with IVC-TT showed

feasibility of SABR for treatment [60]. A larger retrospective multi-institutional experience with 15 patients reported a response rate of 58% after SABR (median dose of 40 Gy in five fractions) for IVC-TT without grade $\geq$ 3 toxicity [61]. The median survival of the cohort was 34 months, and metastatic disease (10/15) and tumor thrombus above the hepatic veins (8/15) was present in the majority of patients. However, independently attributing local control to SABR in this series is difficult given that most patients also received systemic therapy. Nonetheless, LC of IVC-TT with SABR is likely, as evident in a separate report from Freifeld et al. that described two cases of SABR alone resulting in reduction in size of IVC-TT after progression on systemic therapy [62]. It is unlikely that large prospective studies will be conducted in the small cohort of RCC patients with IVC-TT. At this time, we recommend consideration of SABR in the indications included in Table 2.

**Table 2.** Indications for SABR for primary local control of inferior vena cava tumor thrombus.

1. Not a candidate for surgery due to perioperative risk, patient preference, or equipoise on benefit of surgery in context of disease burden.
And
2. One or more of:

1.  Symptomatic disease related to IVC-TT.
2.  Impending symptoms or complications related to IVC-TT (i.e., a bulky thrombus or one that extends close to the hepatic veins or higher).
3.  Locally recurrent IVC-TT following prior IVC thrombectomy.
4.  Not a candidate for systemic therapy or IVC-TT progression on systemic therapy.

### 4.2. Neoadjuvant SABR for RCC with IVC-TT

There is sound rationale to consider neoadjuvant SABR for IVC-TT for improving disease outcome. RCC patients with IVC-TT who underwent definitive surgery are at high risk of subsequent relapse and metastasis, with possible mechanisms including positive surgical margins at the IVC wall leading to local recurrence and IVC-TT generating tumor emboli that cause metastatic spread [59,63]. With this in mind, neoadjuvant SABR is being explored as an approach to enhance disease control in these patients.

Neoadjuvant SABR to RCC IVC-TT is also currently under investigation in an open clinical trial (NCT02473536) [64]. Margulis et al. reported the viability of this strategy in the safety lead-in stage of the above phase II trial with six patients who received SABR to the IVC-TT (40 Gy in 5 fractions) followed by radical nephrectomy and thrombectomy. Of the 81 adverse events reported in the first 90 days after surgery, there were no grade 4/5 toxicities and there was a grade 3 toxicity rate of only 4% [50]. Despite these encouraging observations, the data to support neoadjuvant SABR are limited by the small cohort and limited follow-up period for assessment of oncologic outcomes. Larger, controlled studies on the effectiveness of this approach will be needed prior to routine adoption.

At the present time, the neoadjuvant SABR approach seems well suited to cases where caval wall invasion is suspected and caval resection is not feasible (Table 3). Significant caval wall invasion in the suprarenal cava can usually be addressed with suprarenal cavectomy and ligation or reconstruction. Beyond this point—such as IVC wall invasion at the hepatic vein or supradiaphragmatic level—the IVC cannot feasibly be replaced, and surgical technique often involves manually debriding all adherent tumor [65]. Visualization of this IVC segment can be difficult even with cardiopulmonary bypass and deep hypothermic arrest. In our experience, the likelihood of residual microscopic disease in these situations is high. Given its safety profile, SABR appears to be a rational way to intensify local therapy in this scenario.

**Table 3.** Indications to consider neoadjuvant SABR prior to nephrectomy and IVC thrombectomy.

1. Clinical trial
2. Suspected IVC wall invasion at level of hepatic veins or supradiaphragmatic IVC
3. When primary local control with SABR is otherwise appropriate and candidacy or patient preference to proceed with surgery remains uncertain

## 5. Metastatic RCC

Metastatic RCC encompasses a wide range of disease aggressiveness that varies both in terms of rates of disease progression and disease burden. The International Metastatic Database Consortium (IMDC), externally validated by Heng et al. and now one of the most widely used risk stratification and prognostic models for mRCC, stratifies patients into three risk groupings—favorable (0 risk factors), intermediate (1–2 risk factors) and poor (3 or more risk factors)—based on the following risk factors:

○ Karnofsky performance < 80%;
○ Neutrophils > upper limit of normal;
○ Corrected calcium > upper limit of normal;
○ Platelets > upper limit of normal;
○ Hemoglobin < lower limit of normal;
○ <1 year from diagnosis to systemic therapy [66].

Management of mRCC involves multimodality management. Patients with a high burden of metastatic disease and multiple IMDC risk criteria are managed with upfront systemic therapy. Treatment of the kidney or metastatic lesions is not performed upfront unless symptomatic. Consolidative treatment of the renal primary or metastatic sites can be considered in the event of significant overall response or oligoprogression.

Newer systemic therapies in the form of targeted therapy or ICIs, either alone or in combination, have improved oncologic outcomes in mRCC. Unfortunately, eventual resistance to systemic therapy remains unavoidable [67]. In the context of this inevitable failure of systemic therapy, there remains a role for SABR as a local metastasis-directed therapy in selected mRCC patients. Patients with oligometastatic (OM) RCC are more frequently considered for upfront cytoreductive nephrectomy, active surveillance, and metastasis-directed therapy, especially to delay systemic therapy. In synchronous OM RCC, the decision to perform cytoreductive nephrectomy depends on the plan for metastatic disease—it is indicated when remaining metastatic disease will be managed with active surveillance or metastasis-directed therapy. For those patients in whom systemic therapy is intended, cytoreductive nephrectomy is reasonable when no other IMDC risk factors are present and the majority of disease burden is in the kidney [68,69]. Patients who can have complete control of metastatic lesions with metastasis-directed therapy are also considered for this approach. About half of patients who undergo cytoreductive nephrectomy with synchronous metastasis-directed therapy or active surveillance will remain off systemic therapy for at least one year [70,71]. Those with OM metachronous recurrences may remain off systemic therapy for much longer periods with metastasis-directed therapy [72].

### 5.1. SABR for Metastatic RCC

Multiple studies have demonstrated the efficacy and safety of SABR in the management of mRCC. One of the largest recent meta-analyses examining SABR in mRCC, SABR-ORCA, included 1602 mRCC patients treated with stereotactic radiation and found a 1-year LC rate of ~90% and low incidence of grade 3/4 toxicity of ~1% for both intracranial and extracranial disease [73]. Furthermore, multiple additional studies have documented the feasibility of combining SABR with targeted therapy or ICIs in patients with mRCC [50,53–60,62].

In studies evaluating the use of targeted therapy with SABR, the response rate has ranged from 16% to 76% in treated lesions, with a CR rate of about 29% [74]. Several other retrospective studies have evaluated safety and outcomes with a combination of

first-line targeted therapy and SABR, the largest of which (reported by Liu et al.) included 190 mRCC patients treated with either a tyrosine kinase inhibitor (TKI) + SABR (45%) or a TKI alone and revealed significantly longer survival in the TKI + SABR group than that of the TKI alone. Importantly, SABR was also found to be safe in this study, with no grade 4 or 5 toxicities and a SABR-related grade 3 toxicity of 5.9% [67].

There has recently been interest in combining SABR with ICIs to enhance antitumor effects. Historically, the response rates with nivolumab were around 25%, and initial phase II single-arm prospective trials including the NIVES study (30 Gy in three fractions to a single lesion) and interleukin-2 study did not meet their endpoints, with an overall response rate of only about 16–17% with SABR [75,76]. Subsequently, the RADVAX RCC study using SABR with dual-ICI therapy (nivolumab and ipilimumab) reported a more robust objective response rate (ORR) of 56% [77]. While these trials primarily aimed at eliciting the abscopal effect of SABR by targeting select metastatic sites, more recent studies are evaluating metastasis-directed therapy to all sites of disease using SABR to potentially improve local control and survival. The recent RAPPORT trial employed total metastatic irradiation to up to five metastatic lesions prior to pembrolizumab and observed an ORR of 63%. This combination of SABR and ICIs provided excellent local control, with a 2-year freedom from local progression of 92%. This combination of modalities also proved to be well tolerated, with no grade 4 or 5 adverse events and a 13% rate of treatment-related grade 3 adverse events [78].

To summarize, currently available evidence suggests that SABR in combination with systemic therapy is safe. However, most of these studies are limited by their single-arm prospective or retrospective nature, small sample size, and relatively short follow-up. Furthermore, the optimal dose and fractionation, as well as how to best select patients that may benefit the most from this combination approach, are still unclear. Future prospective randomized trials are warranted to better determine the benefit and utility of a combined SABR and ICIs/targeted therapy approach before it can be implemented in routine clinical practice.

### 5.2. SABR for Brain and Spinal Metastases

Brain metastases (BMs) have been reported in up to 17% of RCC patients. Although patients with active RCC BMs are often excluded from clinical trials using targeted therapy and ICIs, recent studies have reported substantial intracranial activity in mRCC patients with BMs treated with cabozantinib (intracranial response rate 55%) and nivolumab (intracranial response rate 12%) [79,80]. Despite these encouraging data, the blood–brain barrier poses a considerable challenge in the management of BMs with systemic therapies, and local therapy with surgery, SRS, and/or whole-brain radiotherapy (WBRT) remains the mainstay of management for these patients. SRS has recently gained favor over WBRT for patients with limited BMs, as stereotactic techniques not only permit a higher dose per fraction to overcome radioresistance, but also spare these patients from neurocognitive toxicities [81], while maintaining excellent LC rates and comparable survival outcomes [60,67–69,82]. Similarly, SABR for the treatment of RCC spinal metastases has shown LC rates of 70–90% with minimal grade 3 toxicity (nausea) [83]. However, the management of spinal epidural metastatic disease compressing the spinal cord requires additional attention and consideration to avoid serious side effects such as myelopathy. In this setting, surgical decompression or de-bulking followed by post-operative SABR to remaining epidural disease has been shown to be an effective management strategy with excellent LC and safety [84,85].

### 5.3. SABR for OM RCC

OM disease has been defined variably across studies but is usually considered as the number of metastatic lesions limited to ≤5 and/or involving single or a limited number of organs. It can be further divided into subcategories based on the risk of distant micro-

metastasis (Table 4). These subcategories can be helpful in evaluating the probability of future progression at distant sites and the speed of progression.

**Table 4.** Subcategories of OM RCC based on risk of distant metastases.

| 1. Metachronous metastases (>1 year after resection of primary renal tumor) | Indolent disease; best prognosis | <ul><li>Active surveillance</li><li>Metastasectomy</li><li>SABR</li><li>Systemic therapy [72,73,86,87]</li></ul> |
|---|---|---|
| 2. IMDC favorable or intermediate risk | High risk of occult micrometastases | <ul><li>Systemic therapy eventually necessary</li><li>Upfront sequential SABR can preserve health-related quality of life and postpone systemic therapy [88,89]</li></ul> |
| 3. IMDC high risk, grade 4 or sarcomatoid component histology | High risk of occult micrometastatic disease with anticipated rapid progression | <ul><li>Upfront systemic therapy</li><li>Consolidation with SABR to OM disease sites [59]</li></ul> |

Multiple recent prospective and retrospective studies have evaluated SABR for OM disease and have demonstrated durable disease control [70,72,73,75,83]. One of the largest multi-center prospective registry trials of OM patients treated with SABR to all OM sites enrolled 1422 patients with several histologies (including 143 mRCC patients) and reported an OS of 94% at 1 year and 85% at 2 years [90]. Similar results were demonstrated by Tang et al. [88] and Hannan et al. [89], who evaluated definitive-intent SABR to all metastatic sites to allow deferral of systemic therapy. Tang et al. evaluated this in a prospective phase 2 clinical trial including 30 RCC patients, of which 57% received SABR for lung metastases (most commonly 50 Gy in four fractions) and reported a 1-year PFS and systemic therapy-free survival of 64% and 82%, respectively, with 6% grade 3 and one grade 4 (hyperglycemia) adverse events [88]. Hannan et al. conducted a similar trial with the primary objective of delaying the start of systemic therapy by >1 year in >60% patients, and met the endpoint with 91.3% 1-year freedom from systemic therapy, with no significant impact on quality of life outcomes [89]. Overall, these results suggest that SABR may delay systemic therapy in select OM RCC patients without decreasing OS. However, studies performed to date share the limitations of single-arm design with no control arm, shorter follow-up, and use of the relatively novel endpoint of freedom from systemic therapy. Future long-term results of the utility of SABR in deferring systemic treatment are pending to shed light on clinical outcomes, safety, and appropriate biomarkers for patient selection. There is a role of SABR to OM disease sites in subcategory 3 (per Table 4) as well, with the rationale of debulking, ablating any therapy-resistant metastases (see oligoprogressive RCC below), or potential synergy with ICIs to harness the abscopal effect.

### 5.4. SABR for Oligoprogressive RCC

Oligoprogression is defined as disease progression limited to a solitary or a few metastatic lesion(s) while all other metastases are stable or responding to the ongoing systemic treatment. Traditionally, a new line of systemic therapy is indicated at time of progression [19,21], but an alternative approach is ablative local therapy with SABR to oligoprogressive site(s) with a goal to delay the next line of systemic therapy. SABR is being increasingly used for this indication. Since subsequent lines of systemic therapy for mRCC are usually associated with shorter PFS intervals and often cause increased toxicity [91], SABR can potentially improve survival outcomes by eliminating resistant metastatic clones [59]. Multiple studies have evaluated SABR in such an oligoprogressive setting, with 1-year LC ranging from 83 to 100% and a median time to change in systemic therapy of ~1 year [74,77,92–95]. Franzese et al. reported the outcomes of 220 oligoprogressive lesions in 116 patients in a retrospective study, and found a median time to change in systemic treatment of 13.9 months [96]. A prospective study by Cheung et al. reported out-

comes with SABR for all oligoprogressive metastases while continuing the same targeted therapy and observed a 1-year LC rate of 93% for the irradiated tumors and OS of 92%, with no reported grade ≥ 3 SABR-related toxicity. On a secondary analysis of this trial, the cumulative incidence of changing systemic therapy was 47% at 1 year, with a median time of 12.6 months [97]. These studies demonstrate the role of SABR in extending ongoing systemic therapy in select oligoprogressive mRCC patients. Although no clear guidelines exist on patient selection, and the tumor biology and number of progressive lesions that may benefit from this approach are still unknown, these studies nevertheless provide a strong basis for further investigation into this approach in the prospective setting.

*5.5. SABR for Primary Site Cytoreduction in mRCC*

In the historic era during which cytokine therapy was the standard of care for mRCC, the SWOG 8949 and EORTC 30,947 randomized clinical trials demonstrated a statistically significant survival benefit with the addition of cytoreductive nephrectomy (CN) in the management of mRCC [98,99]. In contrast, in the more recent targeted therapy era, analysis of the CARMENA trial revealed that systemic therapy alone with sunitinib was non-inferior in terms of survival compared to CN followed by sunitinib, calling into question the utility of CN in the management of mRCC [100]. However, the role of CN in the current ICI era remains unclear and is being explored in the SWOG S1931 (PROBE trial, NCT 04510597) and Cyto-KIK (NCT04322955) prospective studies. Analogously, the role of SABR as a cytoreductive local therapy to the primary tumor in mRCC in the setting of ICIs is being evaluated in the CYTOSHRINK (NCT04090710) and NRG-GU012 SAMURAI prospective randomized trials [14]. Table 5 describes a summary of potential indications of SABR for RCCs.

**Table 5.** Summary of indications for SABR.

| | |
|---|---|
| Primary localized or locally advanced RCC | • Poor surgical candidates (medically inoperable due to perioperative risk or patient preference) <br> • Large, locally advanced disease <br> • Presence of progressive, recurrent, or symptomatic IVC-TT |
| Metastatic RCC | • Upfront for limited number of metastatic lesions ≤ 5, and/or involving single or limited number of organs (oligometastatic disease) <br> • Consolidation of persistent metastatic sites after systemic therapy (oligo-persistent disease) <br> • Progression of solitary or a few metastatic lesion(s) (oligoprogressive disease) <br> • Primary site cyto-reduction (for bulky or symptomatic disease) |

## 6. Conclusions

In appropriately selected patients, SABR is associated with good oncologic outcomes and a favorable toxicity profile in the management of the full spectrum of RCC presentations, including localized, locally advanced, oligometastatic, oligoprogressive, and widely metastatic disease. Multimodality approaches to localized RCC, including SABR in the operable setting, are being investigated. In the setting of mRCC, SABR alone or in combination with systemic regimens has demonstrated encouraging clinical outcomes with modest toxicity. Other potential applications of SABR, including neoadjuvant therapy for RCC with IVC-TT, SABR in OM RCC to defer initiation of systemic therapy, SABR in oligoprogressive RCC to delay a change in systemic regimen, and cytoreduction of primary disease in mRCC are evolving and should be further explored in prospective clinical trials.

**Author Contributions:** R.K.R., R.U. and S.-J.W. contributed to the study design, screened studies for eligibility, extracted the data, and wrote the paper. E.A.S. contributed to the study design and helped write the paper. S.D. contributed to the study design and helped write the paper. All authors have read and agreed to the published version of the manuscript.

**Funding:** This research is supported by a grant from the National Cancer Institute (2P30CA016058).

**Conflicts of Interest:** The authors declare no conflict of interest.

**Abbreviations**

| | |
|---|---|
| RCC | Renal Cell Carcinoma |
| mRCC | Metastatic Renal Cell Carcinoma |
| ICIs | Immune Checkpoint Inhibitors |
| TKIs | Tyrosine Kinase Inhibitors |
| SABR | Stereotactic Ablative Radiation Therapy |
| RT | Radiation Therapy |
| SRS | Stereotactic Radiosurgery |
| IVC-TT | Inferior Vena Cava–Tumor Thrombus |
| SPECT | Single-photon Emission Computed Tomography |
| CT | Computed Tomography |
| MRI | Magnetic Resonance Imaging |
| DISSRM | Delayed Intervention and Surveillance for Small Renal Masses Registry |
| IROCK | International Radiosurgery Consortium of the Kidney |
| IMDC | International Metastatic Database Consortium |
| LC | Local Control |
| PFS | Progression-free Survival |
| OS | Overall Survival |
| CSS | Case-specific Survival |
| HR | Hazard Ratio |
| CIs | Confidence Intervals |
| BED | Biological Effective Dose |
| CR | Complete Response |
| eGFR | estimated Glomerular Filtration Rate |
| OM | Oligometastatic Disease |
| OP | Oligoprogressive Disease |
| BM | Brain Metastases |
| WBRT | Whole-Brain Radiotherapy |

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
