# Peer review of "Incorporating Stereotactic Ablative Radiotherapy into the Multidisciplinary Management of Renal Cell Carcinoma"

_curroncol, doi:10.3390/curroncol30120749_

Round 1

Reviewer 1 Report

Comments and Suggestions for Authors

This reviewer does not have any major comments and the article is suitable for publication; however, a couple of minor modifications are highly recommended.

Line 29: “Other subtypes” can be changed to “Major subtypes”.

Lines 32-34. Median value is only one value. This sentence could be rewritten to provide the median value followed by range and IQR.

Author Response

  1. This reviewer does not have any major comments and the article is suitable for publication; however, a couple of minor modifications are highly recommended.

Line 29: “Other subtypes” can be changed to “Major subtypes”.

Thanks a lot for this comment. We have added this change, Line 29 now reads “Major subtypes of RCC include clear cell (75-85%), papillary (10-15%) and chromophobe (5-10%) tumors.”

  1. Lines 32-34. Median value is only one value. This sentence could be rewritten to provide the median value followed by range and IQR.

Thank you for the comment. This was a range of median values from several prior studies. We have modified the language of the sentence, and it now reads as “Overall survival across several prior studies ranges from 6 to 12 months in these patients with metastatic RCC”.

Reviewer 2 Report

Comments and Suggestions for Authors

Same table with the main studies on the treatment of primary tumours and oligometastases would improve the value of the publication.

The aim of this article is to present the role of radiotherapy in the multidisciplinary treatment of renal cancer.

In contrast to previous review articles which give a partial review of the different indications, this article analyses comprehensively all existing indications .In my opinion, it is the most comprehensive article in this field.

This article highlights the indications for a therapeutic approach not often used in a tumour considered to be radioresistant and whose treatment, the SABR presents an effective alternative.

I believe that presenting a couple of additional tables with the most frequent therapeutic indications (localised/locally advanced tumours and oligometastatic tumours) may improve the reader's understanding and the scientific value of the article submitted.

The bibliography is up to date.

Author Response

  1. Same table with the main studies on the treatment of primary tumours and oligometastases would improve the value of the publication.

The aim of this article is to present the role of radiotherapy in the multidisciplinary treatment of renal cancer.
In contrast to previous review articles which give a partial review of the different indications, this article analyses comprehensively all existing indications .In my opinion, it is the most comprehensive article in this field.
This article highlights the indications for a therapeutic approach not often used in a tumour considered to be radioresistant and whose treatment, the SABR presents an effective alternative.
The bibliography is up to date.

Thanks a lot for the comment, we appreciate the positive response. We have added a table (Table 1) summarizing the studies describing outcomes of patients with localized RCCs treated with SABR. Summary of the studies evaluating oligometastatic / oligoprogressive settings have been recently published (doi: 10.1016/j.eururo.2022.06.017; PMID: 35843777) and would be beyond the scope of this review.

  1. I believe that presenting a couple of additional tables with the most frequent therapeutic indications (localised/locally advanced tumours and oligometastatic tumours) may improve the reader's understanding and the scientific value of the article submitted.

Thank you for the suggestion. We have added a table (Table 5) summarizing the indications for SABR in the oligometastatic / oligoprogressive settings. Table 2 and Table 3 describe indications of SABR in the locally advanced and neoadjuvant settings, respectively. 

Reviewer 3 Report

Comments and Suggestions for Authors

Dear Authors, I have read with interest your manuscript. The paper addresses an very interesting issue regardingrenal carcinomas, one of the most common type of kidney cancer in adults, responsible for approximately 90–95% of cases, but also one of the cancers most strongly associated with paraneoplastic syndromes, most often due to ectopic hormone production by the tumour.

I would like to address a few suggestions/ questions:

The purpose of your review was not clearly explained and I think that the methods were insufficiently described. For this purpose, more details regarding the studies you included in your review should be given. Please state very clearly in a few phrases the aim of this review.

.

Also, bring details about when the information your review was based on was published and when you have accessed it.

You presented the information very well and it was very well structured

How can your review bring improvements in this field? What can be put into practice, thus being able to better select which patients would benefit from a better management of renal call carcinomas?

Author Response

  1. Dear Authors, I have read with interest your manuscript. The paper addresses an very interesting issue regarding renal carcinomas, one of the most common type of kidney cancer in adults, responsible for approximately 90–95% of cases, but also one of the cancers most strongly associated with paraneoplastic syndromes, most often due to ectopic hormone production by the tumour.
    I would like to address a few suggestions/ questions:
    The purpose of your review was not clearly explained and I think that the methods were insufficiently described. For this purpose, more details regarding the studies you included in your review should be given. Please state very clearly in a few phrases the aim of this review.

Thank you for the thorough review. We have added further description on the inclusion and exclusion criteria and aims of the review in the Evidence acquisition section, page 2, paragraph 4 which now reads Any prospective or retrospective studies and review articles describes results with the use of SABR for RCC, with original article in English language or an available translated version, were included. Preclinical and modeling studies with clinical information, and papers with incomplete, missing, or duplicated data were included. We did not require a minimum median follow-up duration for inclusion. The major objective of this descriptive review was to summarize the current literature supporting the use of SABR in the multidisciplinary management of localized and metastatic RCC, in combination with surgery and systemic therapy. We also aimed at highlighting the current knowledge gaps and the need for future prospective studies”.

  1. Also, bring details about when the information your review was based on was published and when you have accessed it.

We have added further clarification in the Methods section, page 2, paragraph 4 which now reads Medical literature including clinical trials, clinical studies, case reports, cohort studies and review articles published from January 2000 to April 2023 were searched in PubMed. The search was performed on May 1, 2023, and did not have a language filter.

  1. You presented the information very well and it was very well structured
    How can your review bring improvements in this field? What can be put into practice, thus being able to better select which patients would benefit from a better management of renal call carcinomas?

Thank you for the comment. We have modified the Conclusions section to include the following statement: “In appropriately selected patients, SABR is associated with good oncologic outcomes and favorable toxicity profile in the management of the full spectrum of RCC presentations, including localized, locally advanced, oligometastatic, oligoprogressive, and widely metastatic disease.”. We have also added a Table describing potential indications for SABR in various settings.

Reviewer 4 Report

Comments and Suggestions for Authors

The paper is well written and logically organized. Would recommend to discuss your findings in the context of organ transplantation. Most donors with localized RCC present with a renal mass noted incidentally on imaging or at procurement and this asks the question: how investigate the nodule/tumor, the issue of margins, the oncological risk of using the donor considering the kidney but also other organs. By a pathologist perspective the opportunity of ablative radiotherapy can help in managment of these such donors. Quote PMID: 24034231, PMID: 32535833

Comments on the Quality of English Language

Well written

Author Response

1. The paper is well written and logically organized. Would recommend to discuss your findings in the context of organ transplantation. Most donors with localized RCC present with a renal mass noted incidentally on imaging or at procurement and this asks the question: how investigate the nodule/tumor, the issue of margins, the oncological risk of using the donor considering the kidney but also other organs. By a pathologist perspective the opportunity of ablative radiotherapy can help in managment of these such donors. Quote PMID: 24034231, PMID: 32535833

Thanks a lot for bringing up this important consideration. Although this may be an indication in the future, to our knowledge, there haven’t been any studies yet evaluating the feasibility and safety of doing SABR on a donor or transplanted kidney. The authors feel that SABR for transplanted kidney may have considerable risks including damage to the transplanted kidney and potentially introducing cancer into the recipient. Radiation is expected to cause at least some injury to the surrounding renal parenchyma, which will likely make the transplantation more difficult. Describing this in the results may be out of scope for the current review.

Reviewer 5 Report

Comments and Suggestions for Authors

A narrative review that reports the stereotactic ablative radiotherapy (SABR) in the management of all stages of renal cell carcinoma (RCC).

I would like to thank the authors for their great work to summarise all aspects of the use of SABR in the management of RCC. The manuscript has been written in good English; the presentation of the subject is clear and well designed. But I have a few minor comments that are listed below:

1. Line 29: In my opinion, this sentence reads little bit confusing: I understand that RCC is a subtype, and there are other subtypes as ccRCC, pRCC, chRCC etc. However, this would be in another way phrased, for example: RCC is the most prevalent histological type of primary renal neoplasms and also carcinomas, which has subtypes such as ccRCC, pRCC, chRCC. Other histological types are oncocytoma, angiomyolipoma ..... which are benign. This is just an example, but I suggest the authors to rephrase this sentence.

2. Some references are given one by one in square brackets, while some have been grouped. I think it would read easier and take more little space if they are all merged in case more than one reference is used in a sentence (such as, [1, 3-5, 13, 18-23].

3. Even though it might not be mandatory for the journal, I suggest the authors to make a list of abbreviations, as it is sometimes not easy to re-find the meaning of an abbreviation if you cannot remember it, especially a lot after it has been for the first time used.

4. Table 1 and 3 are presented as divided in separate pages, which makes it difficult to read. It might be due to page outlay, but I think it would be better if they are printed on a single page.

5. The abbreviation "TT" has been used for two definitions: once for tumor thrombus, and then for targeted therapy. Can the authors please change one of these identical abbreviations?

6. Maybe not important, but I think the abbreviation IO for immunotherapy is something not very commonly used. May it be replaced with another more commonly used abbreviation?

Author Response

A narrative review that reports the stereotactic ablative radiotherapy (SABR) in the management of all stages of renal cell carcinoma (RCC).
I would like to thank the authors for their great work to summarise all aspects of the use of SABR in the management of RCC. The manuscript has been written in good English; the presentation of the subject is clear and well designed. But I have a few minor comments that are listed below:

  1. Line 29: In my opinion, this sentence reads little bit confusing: I understand that RCC is a subtype, and there are other subtypes as ccRCC, pRCC, chRCC etc. However, this would be in another way phrased, for example: RCC is the most prevalent histological type of primary renal neoplasms and also carcinomas, which has subtypes such as ccRCC, pRCC, chRCC. Other histological types are oncocytoma, angiomyolipoma ..... which are benign. This is just an example, but I suggest the authors to rephrase this sentence.

Thank you for the comment and the thorough review. We have modified this line in the manuscript. Line 29 now reads “Major subtypes of RCC include clear cell (75-85%), papillary (10-15%) and chromophobe (5-10%) tumors. Other histological types of renal neoplasms are oncocytoma and an-giomyolipoma (3-5%).”

  1. Some references are given one by one in square brackets, while some have been grouped. I think it would read easier and take more little space if they are all merged in case more than one reference is used in a sentence (such as, [1, 3-5, 13, 18-23].

Thank you for the comment. We have made the suggested changes in the formatting of references.

  1. Even though it might not be mandatory for the journal, I suggest the authors to make a list of abbreviations, as it is sometimes not easy to re-find the meaning of an abbreviation if you cannot remember it, especially a lot after it has been for the first time used.

Thank you for the comment. We have added a list of abbreviations used at the end of the article – this can be suitably placed by the editors after the abstract and before the full paper.

  1. Table 1 and 3 are presented as divided in separate pages, which makes it difficult to read. It might be due to page outlay, but I think it would be better if they are printed on a single page.

Thank you for the comment. We have modified the formatting such that all the tables are on the same page.

  1. The abbreviation "TT" has been used for two definitions: once for tumor thrombus, and then for targeted therapy. Can the authors please change one of these identical abbreviations?

Thank you for the catch. We have kept TT to imply for tumor thrombus, while we have expanded TT as targeted therapy at all occurrences.

  1. Maybe not important, but I think the abbreviation IO for immunotherapy is something not very commonly used. May it be replaced with another more commonly used abbreviation?

Thanks for the suggestion. We have modified “IO” to “Immune checkpoint inhibition (ICI)” at all occurrences.
